# The Association between the EAT–Lancet Diet and Diabetes: A Systematic Review

**DOI:** 10.3390/nu15204462

**Published:** 2023-10-21

**Authors:** Xiaoxiao Lin, Shuai Wang, Jinyu Huang

**Affiliations:** Affiliated Hangzhou First People’s Hospital, Zhejiang University School of Medicine, Zhejiang University, Hangzhou 310030, China

**Keywords:** EAT–Lancet diet, diabetes, association, benefits, systematic review

## Abstract

Several studies have explored the association between diabetes and the EAT–Lancet diet. Thus, the objective of our study was to conduct a systematic review to analyze and summarize all clinical studies concerning the association between diabetes and the EAT–Lancet diet. We undertook a comprehensive search of the Embase, Cochrane, and PubMed databases up to 15 August 2023. All clinical studies concerning the association between diabetes and the EAT–Lancet diet were summarized and analyzed. In total, our systematic review included five studies of four prospective studies and one cross-sectional study, encompassing 259,315 participants. All the included studies were evaluated as high quality. The outcomes from all studies indicated that adherence to the EAT–Lancet diet was correlated with a reduced risk of diabetes. In conclusion, the EAT–Lancet diet may be an effective dietary intervention for diabetes. Nevertheless, the number of studies examining the association between diabetes and the EAT–Lancet diet is limited. Further high-quality studies are required to expand our understanding of the benefits of the EAT–Lancet diet for patients with diabetes.

## 1. Introduction

Diabetes mellitus represents a complex set of endocrine system diseases characterized by abnormally high blood glucose levels. These include type 1 diabetes mellitus (T1DM) [1], type 2 diabetes mellitus (T2DM) [2], and other specific types like gestational diabetes [3]. Current estimates suggest that the global prevalence of diabetes stands at 6.1% [4]. This condition has emerged as a significant public health concern. Management strategies for diabetes encompass lifestyle interventions such as exercise and dietary modifications, as well as pharmaceutical treatments. Central to lifestyle management is the emphasis on a balanced diet. Previous studies have demonstrated that some dietary patterns play an important role in prevention and management of diabetes [5,6,7,8,9,10,11]. Several dietary patterns have been advocated for individuals with diabetes [12,13], including calorie restriction (CR) [14], intermittent fasting [15] or alternative-day fasting (ADF), time-restricted eating (TRE) [16], and the 5:2 diet, and specific dietary compositions like the EAT–Lancet diet [17,18,19], low and very low carbohydrate diets [20], ketogenic diets [9], and the Mediterranean diet [21]. Among these, the EAT–Lancet diet has garnered increasing attention, and was introduced by the EAT–Lancet Commission in 2019 as a blueprint for healthy diets [22,23]. The EAT–Lancet diet is the result of a comprehensive study conducted by the EAT–Lancet Commission, a collaboration between 37 experts from 16 countries that included nutritionists, agriculturalists, ecologists, and more [23]. Their aim was to determine a sustainable diet that would optimize both human health and environmental sustainability. The resulting EAT–Lancet diet, sometimes referred to as the “planetary health diet,” is characterized by several key recommendations [22,23,24,25,26], including the following. High intake of plant-based foods: the diet prioritizes the consumption of whole grains, fruits, vegetables, legumes, and nuts. Limited intake of animal-based foods: it suggests a significant reduction in the consumption of red meat and other animal-based products. Fish and poultry are allowed in moderate amounts. Limited intake of added sugars and refined grains: added sugars and refined grains should be minimized. The diet recommends less than 31 g of added sugar and roughly 232 g of total starchy vegetables (like potatoes) and whole grains combined per day. Healthy fats: while the diet is moderate in total fat, it prioritizes unsaturated fats like those found in avocados, nuts, and certain oils over saturated and trans fats. Limit dairy: the diet suggests a modest intake of dairy, mainly from low-fat sources. Nutritional targets: the diet aims to ensure that people receive all the essential nutrients they require for good health. Environmental sustainability: beyond its nutritional aspects, the EAT–Lancet diet takes into account the environmental impact of food production. The recommendations seek to mitigate the negative impacts of agriculture on climate change, biodiversity loss, land and water use, and nutrient pollution. Flexibility: recognizing that global dietary changes need to be adaptable to local cultures, economies, and environments, the EAT–Lancet diet is designed to be flexible. The core principles can be applied in various ways across different global contexts. The Commission posits that widespread adoption of the EAT–Lancet diet would result in better health outcomes for individuals and lead to a more sustainable and resilient food system that can support a growing global population without exhausting the Earth’s natural resources. The diet is a response to the two challenges of global malnutrition (both undernutrition and obesity-related health issues like diabetes) and unsustainable agricultural practices that threaten planetary health. The specific macronutrient distribution of the EAT–Lancet diet is roughly around 50–55% carbohydrates, 29–35% fats, and 15–25% proteins, based on daily total caloric intake of around 2500 kcal/day for men and 2000 kcal/day for women [23]. The components of the EAT–Lancet diet is outlined in Table 1.

The EAT–Lancet diet has some potential beneficial effects for health-related parameters such as cognitive function, metabolic health, and cardiovascular health [23,25]. The nutrient-dense components of the EAT–Lancet diet are essential for cognitive health, providing antioxidants and omega-3 fatty acids that combat processes impairing brain function. The diet’s high fiber content stabilizes blood sugar levels, contributing to optimal metabolic health, and aids in the prevention of conditions like insulin resistance and diabetes. Moreover, it supports cardiovascular health by promoting the intake of unsaturated fats and reducing foods high in saturated fats, helping manage blood pressure, cholesterol levels, and overall heart and artery health, thereby lowering the risks of heart disease and stroke.

The EAT–Lancet diet has been designed for the general population, but several studies have found that the EAT–Lancet diet is associated with a lower risk of developing diabetes. For instance, Langmann et al. [27] explored the relationship between the EAT–Lancet diet and the risk of T2DM, concluding that adherence to this diet was linked to a decreased risk of T2DM. López et al. [17] examined the relationship between the EAT–Lancet healthy reference diet (EAT–HRD) and T2DM incidence, and found that compliance with legume, fish, and red meat recommendations was associated with a reduced incidence of T2DM. A recent prospective study [19] also indicated that adherence to the EAT–Lancet reference diet was correlated with a decreased T2DM risk across all levels of genetic susceptibility. However, no systematic review has been undertaken to summarize these studies. Hence, the objective of our research is to conduct a systematic review to analyze and summarize all clinical studies examining the association between the EAT–Lancet diet and diabetes.

## 2. Methods

### 2.1. Search Strategy

We conducted this systematic review in accordance with the Cochrane Collaboration guidelines. The protocol for our systematic review was predetermined and has been registered with the INPLASY website (ID:202380068). The findings are reported in line with the PRISMA checklist [28]. A comprehensive search was undertaken in the databases of Embase, PubMed, and Cochrane up to August 15, 2023. Our search included the following terms: (“eat-lancet diet” OR EAT-Lancet OR “Planetary Health Diet” OR “plant diet” OR “EAT-Lancet healthy reference diet” OR “plant-based diet” OR EAT-HRD OR “vegetarian diet” OR EAT-LDP OR “EAT-Lancet reference diet” OR “EAT-Lancet diet pattern”) AND (diabetes OR diabetic OR “Non-Insulin-Dependent Diabetes Mellitus” OR “type 2 diabetes” OR T2DM OR “type 1 diabetes” OR T1DM OR “Adult-Onset Diabetes Mellitus” OR “Diabetes Mellitus, Slow-Onset” OR “Diabetes Mellitus, Type II” OR “Noninsulin-Dependent Diabetes Mellitus” OR “Insulin Dependent Diabetes Mellitus” OR “Insulin-Dependent Diabetes Mellitus” OR “gestational diabetes mellitus” OR “Diabetes Mellitus, Gestational” OR “Pregnancy-Induced Diabetes” OR “Diabetes, Pregnancy Induced” OR “diabetes mellitus” OR “Noninsulin Dependent Diabetes Mellitus”). We also reviewed the references of pertinent reviews. The search was collaboratively performed by two authors (LXX and WS). In cases of uncertainty, a discussion was held with a third author (HJY).

### 2.2. Inclusion Criteria

Utilizing the PICOS framework, we established the following inclusion criteria. (P) Population: adults aged above 18 years. (I) Intervention: EAT–Lancet diet. (C) Control: individuals not adhering to the EAT–Lancet diet. (O) Outcomes: the association between diabetes and adherence to the EAT–Lancet diet. (S) Study types: clinical studies including cross-sectional studies, case-control studies, cohort studies, and randomized controlled trials (RCTs). We excluded editorials, duplicates, commentaries, conference abstracts, supplements, and case reports.

### 2.3. Quality Appraisal and Data Extraction

Different methodologies were utilized to assess the quality of studies, depending on their respective designs. The quality of cohort and case-control studies was determined using the Newcastle–Ottawa Scale (NOS), consisting of eight critical questions targeting participant selection, group comparability, and exposure verification [29]. The cross-sectional studies were examined through the AHRQ checklist’s lens [30], comprising 11 specific criteria. To ensure clarity in presentation and consistency in outcomes, scores from these assessments were classified into three distinct quality levels: low, moderate, and high. The task of data extraction was carried out independently by two authors (LXX and WS). They organized the information into two structured tables: the first encompassed details such as the study, year, type of study, sample size, average age, percentage of females, database used, questionnaires employed, types of diabetes, and duration of follow-up. The second table focused on the principal findings and the outcome of the quality evaluation.

## 3. Results

### 3.1. Literature Search

Our initial database search retrieved a total of 1816 records. After deduplicating, 1397 records remained for screening through titles, abstracts, and full texts. Ultimately, five studies comprising four prospective studies [17,18,19,27] and one cross-sectional study [31] were included in the systematic review. The progression of the search and selection process is depicted in Figure 1.

### 3.2. Study and Patient Characteristics, and the Assessment of Quality

A total of 259,315 participants were included in the studies. The individual study sample sizes varied, ranging from 24,494 to 74,671 participants. The proportion of female participants in the studies spanned from 52.6% to 100%. The mean age ranged from 41.2 to 58.1 years. Type 2 diabetes incidence was evaluated in four studies, and one study assessed all types of diabetes. The time of follow-up ranged from 2.2 years to 24.3 years.

The data for these studies were extracted from five distinct databases: the Danish Diet, Cancer and Health cohort study; Malmö Diet and Cancer (MDC) study; UK Biobank; EPIC-Oxford study; and the Mexican Teachers’ Cohort (MTC). For the quality assessment, the cross-sectional study scored 10 on the AHRQ scale [31], and the four cohort studies scored 9 on the NOS scale [17,18,19,27]. All the studies were classified as high quality. A comprehensive summary of study characteristics is shown in Table 2, and their main findings and quality assessments are summarized in Table 3.

### 3.3. The Relationship between EAT–Lancet Diet and Diabetes

Langmann et al. [27] explored the relationship between adherence to the EAT–Lancet diet and the risk of T2DM among 54,232 participants in the Danish Diet, Cancer and Health cohort study. Within this cohort of middle-aged Danish adults, a stronger adherence to the EAT–Lancet dietary guidelines was correlated with a decreased risk of T2DM onset. This indicates a potential public health benefit, suggesting that following this diet could play a preventive role against T2DM. Additionally, when comparing the EAT–Lancet diet score with another dietary evaluation metric, the Alternative Healthy Eating Index 2010 (AHEI-2010), the associations regarding T2DM risk displayed comparable magnitudes and trends. The study’s conclusion emphasized that high scores on both the EAT–Lancet diet and AHEI-2010 were associated with a lower risk of T2DM among middle-aged Danes. This association persisted after adjusting for potential confounders. These findings underscore that diets centered on environmental sustainability principles offer diabetes risk reduction benefits akin to those of other recognized health-focused diets. In light of these findings, it is advisable for subsequent cohort studies to delve into the adherence to the EAT–Lancet diet, evaluating its connection with long-term health outcomes in varied global populations.

Zhang et al. [19] conducted a prospective cohort study to investigate the association between the EAT–Lancet diet and T2DM among 24,494 adults from the Malmö Diet and Cancer (MDC) study. They observed that higher adherence to the EAT–Lancet diet index (EAT–LDI) was linked to a reduced risk of T2DM, demonstrating a dose–response relationship. The EAT–LDI ranges from 0 to 42 points, with a score of 3 for the highest adherence and a score of 0 for the lowest for 14 components in the EAT–Lancet diet. This correlation persisted even when accounting for participants’ genetic predisposition to T2DM, attesting to its strength. The researchers further confirmed the reliability of this association through multiple sensitivity analyses. Based on their data, they estimated that if all study participants maintained an adherence score of ≥23 points on the EAT–LDI, roughly 12.9% of T2DM cases could potentially be averted. They concluded that adherence to the EAT–LDI, which reflects the principles of the EAT–Lancet reference diet (EAT–LRD), could significantly reduce T2DM risk across varying genetic susceptibilities. These findings bolster the EAT–Lancet Commission’s recommendations, emphasizing the benefits of a sustainable diet.

Xu et al. [18] examined the relationship between the EAT–Lancet diet pattern (EAT–LDP) and the occurrence of T2DM within a cohort of 59,849 adults sourced from the UK Biobank, all of whom did not have a prior diabetes diagnosis. Over a median tracking period of 10 years, 2461 participants developed T2DM. For each incremental point increase in the diet score, there was a corresponding 6% decrease in the risk of T2DM. Furthermore, they discerned a substantial indirect connection between the EAT–LDP score and T2DM, indicating that approximately 44% of the relationship between them was mediated by body mass index (BMI). Additionally, they noted that 40% of the link between them was mediated by waist circumference. The outcomes of their research unequivocally demonstrated that greater adherence to the EAT–LDP was related to a diminished risk of developing T2DM over an extended period. Their conclusion underscores the clinical relevance of these findings, particularly in the context of the escalating global burden of diabetes. Notably, their results suggest that the EAT–LDP represents an attainable and sustainable goal that should be actively promoted for diabetes prevention efforts.

Knuppel et al. [31] undertook an investigation into the relationship between the EAT–Lancet score and the occurrence of diabetes within a cohort comprising 46,069 participants from the EPIC-Oxford study. Their findings revealed a significant outcome that a high level of adherence to the EAT–Lancet score was strongly linked to a lower risk of diabetes. They concluded that the EAT–Lancet reference diet exhibited favorable associations concerning diabetes within this extensive prospective cohort consisting of British adults. Moreover, their research also brought to light additional noteworthy insights. Specifically, they identified that the EAT–Lancet reference diet displayed advantageous associations not only with diabetes but also with ischemic heart disease within the study population.

López et al. [17] conducted a comprehensive study aimed at investigating the potential link between adherence to the EAT–HRD and the incidence of T2DM among 74,671 women drawn from the Mexican Teachers’ Cohort (MTC). Within this particular group of Mexican women, the researchers made an intriguing discovery that there was a protective association between higher adherence to the EAT–Lancet score and the incidence of T2DM. However, it is important to note that this association exhibited some degree of imprecision when comparing those with higher adherence to the EAT–HRD score with those with lower adherence. Specifically, women who adhered to the fish, red meat, and legume recommendations experienced a decreased incidence of T2DM. Surprisingly, contrary to initial expectations, meeting the recommended limit for added sugars (<31 g/d) was related to a higher incidence of T2DM. In conclusion, the research findings suggest that a greater commitment to the EAT–HRD score may offer a means of preventing T2DM. Additionally, meeting fish, red meat, and legume recommendations appears to be particularly beneficial in reducing the occurrence of T2DM.

## 4. Discussion

To the extent of our knowledge, this is the first systematic review to analyze and summarize all clinical studies pertaining to the relationship between diabetes and the EAT–Lancet diet. In total, five studies comprising four prospective studies and one cross-sectional study and 259315 participants were included in the systematic review. All studies were evaluated as high quality, and all studies demonstrated that adhere to the EAT–Lancet diet was related to a low risk of diabetes.

The EAT–Lancet diet represents an emergent universal health reference diet, designed to serve as a foundation for evaluating both the health and environmental implications of transitioning from prevalent standard diets, which are characterized by some content of unhealthy foods. Previous studies have explored the effects of the EAT–Lancet diet for several diseases, including diabetes [19,31], cardiovascular diseases (CVDs) [32,33,34], stroke [35], and cancer [36]. For example, in a separate study by Colizzi et al. [33], researchers devised a diet score centered on the EAT–Lancet diet and examined its connection with cardiovascular incidents and ecological footprints, analyzing data from 35,496 individuals in the EPIC-NL study. The results indicated that participants who closely followed the EAT–Lancet diet reported fewer instances of cardiovascular ailments (14% reduction), coronary artery disease (12% reduction), and strokes (11% reduction). From an environmental perspective, staunch adherence to the EAT–Lancet diet was associated with diminished greenhouse gas emissions, land utilization, freshwater and marine eutrophication, and soil acidification. A recent analysis [36] delved into the correlation between following the EAT–Lancet diet and the occurrence and death rates of lung cancer, utilizing information from the Prostate, Lung, Colorectal, and Ovarian (PLCO) trial involving 101,755 American adults. While past studies underscored the diet’s potential in diminishing chronic disease risks and overall death rates, its direct link to lung cancer remained ambiguous. The findings revealed that optimal compliance with the EAT–Lancet diet, as reflected by higher scores, correlated with a marked reduction in lung cancer cases and death rates, especially in non-small-cell lung cancer instances. This suggests that strict adherence to the diet could contribute to lowering lung cancer risks. In our study, we conducted a systematic review to summarize and analyze all clinical studies of the EAT–Lancet diet for diabetes. In 2019, Knuppel et al. [31] constructed an EAT–Lancet score based on the 14 components of the EAT–Lancet diet recommendations, with the scores ranging from 0 to 14. This was the first study to explore the association between the EAT–Lancet diet and diabetes. Three subsequent studies [17,18,27] adopted this binary scoring criterion for constructing their EAT–Lancet diet score. Recently, Zhang et al. [19] introduced a newly developed EAT–Lancet score with a gradual scoring criterion. All these findings provide the evidence that aids in comprehending the health implications of the EAT–Lancet reference diet in relation to a lower risk of T2DM, which is consistent with other plant-based dietary patterns [37,38,39,40,41,42,43,44]. Various biological mechanisms may underlie the advantages of the EAT–Lancet diet in preventing diabetes [17,23,26]. These mechanisms include enhanced postprandial glucose regulation, improved glycemic control, heightened insulin sensitivity, decreased intake of harmful components, and reduced chronic inflammation. In our systematic review, all included studies demonstrated that the EAT–Lancet diet was related to a lower risk of developing diabetes, especially for type 2 diabetes. The benefits may be associated with the components of the EAT–Lancet diet [23], emphasizing fruits, vegetables, whole grains, legumes, nuts, and unsaturated oils, with minimal red meat, processed meat, and sugar, which aligns with the key principles for managing and preventing diabetes. Rich in high-fiber, nutrient-dense foods, the diet supports glycemic control, insulin sensitivity, and weight management, which are all essential factors in diabetes care. Moreover, its low allowance for foods linked to increased diabetes risk, such as red and processed meats and refined carbohydrates, further reinforces its potential benefits.

For the components comprising the EAT–Lancet diet, individuals who adhered to fish, red meat, and legume recommendations exhibited a decreased incidence of T2DM, in line with findings from prior research. López et al. [17] demonstrated that abiding by the recommendation of restricting red meat consumption to 14 g or less per day was related to a reduced risk of T2DM. Other cohort studies have also yielded evidence indicating that diminishing the intake of red meat can reduce the incidence of T2DM [37,42]. It is noteworthy that meat production stands as the foremost contributor to methane emissions and livestock production. Hence, reducing global red meat consumption assumes paramount importance in the pursuit of environmental sustainability goals. Similarly, the consumption of fish was identified as a protective factor against the incidence of T2DM. However, their study revealed that the dairy consumption recommendation was linked to an elevated risk of T2DM, which has garnered considerable attention in previous studies [45,46,47]. Notably, the EAT–HRD score encompasses both low- and high-fat dairy components, potentially elucidating the observed increase in T2DM incidence within their study population due to dairy consumption. In addition, they observed that adhering to the limiting added sugar intake to 31 g per day recommendation was associated with an increased risk of T2DM, which was consistent with the findings of Xu’s study [18], which reported an 85% higher T2DM rate among those adhering to this recommendation. It seems that the relationship between diabetes and the EAT–Lancet diet is affected by the different components. However, it should be noted that, in the recent study conducted by Zhang et al. [19], they did not conduct a detailed analysis of the components. Instead, they employed an iterative approach to recalculate the EAT–LDI scores by excluding each component. Subsequently, they evaluated the relationship between the recalculated scores and the risk of T2DM. Their findings demonstrated that the single components only slightly influenced the inverse relationship between the risk of T2DM and the overall score. More studies are needed to explore the association between the components of the EAT–Lancet diet and diabetes. In addition, Zhang et al. demonstrated that a high level of adherence to the EAT–LDI score was correlated with a reduced risk of T2DM across different individuals with varying levels of genetic susceptibility, and their study underscores the overall healthiness of the EAT–Lancet diet for the general population, no matter what their genetic predisposition to T2DM is. It is worth noting that the polygenic risk score (PRS) used for identifying individuals at an elevated risk of T2DM comprises 116 SNPs that encompass diverse metabolic pathways or even those with unidentified functions [48]. This finding showed that the interaction between T2DM-PRS and the EAT–Lancet diet may indeed be exceedingly modest and possibly elusive when employing conventional methodologies, possibly confounded by other variables [49]. Additionally, recent research has indicated that distinct genetic backgrounds underlie different subtypes of T2DM [50]. Hence, forthcoming studies should be conducted to provide a more comprehensive understanding.

There are some limitations in the systematic review. Firstly, due to the heterogeneity with regard to the different grading score, a meta-analysis could not be conducted. Secondly, there are five studies exploring the association between the EAT–Lancet diet and diabetes. The number of studies about this topic is limited, and, among these studies, 4 out of 5 studies indicate the beneficial effect of the EAT–Lancet diet on the risk of types 2 diabetes, and one study demonstrates that the EAT–Lancet diet is related to the risk of diabetes. The association between the EAT–Lancet diet and type 1 diabetes should be investigated in further studies. In addition, there are no RCTs to explore the association between diabetes and the EAT–Lancet diet. More high-quality studies with strict design, long-term intervention and follow-up are needed.

In conclusion, the EAT–Lancet diet may be an effective dietary intervention for diabetes. Nevertheless, the number of studies examining the association between diabetes and the EAT–Lancet diet is limited. As such, further high-quality studies are required to expand our understanding of the benefits of the EAT–Lancet diet for patients with diabetes.

## Figures and Tables

**Figure 1 nutrients-15-04462-f001:**
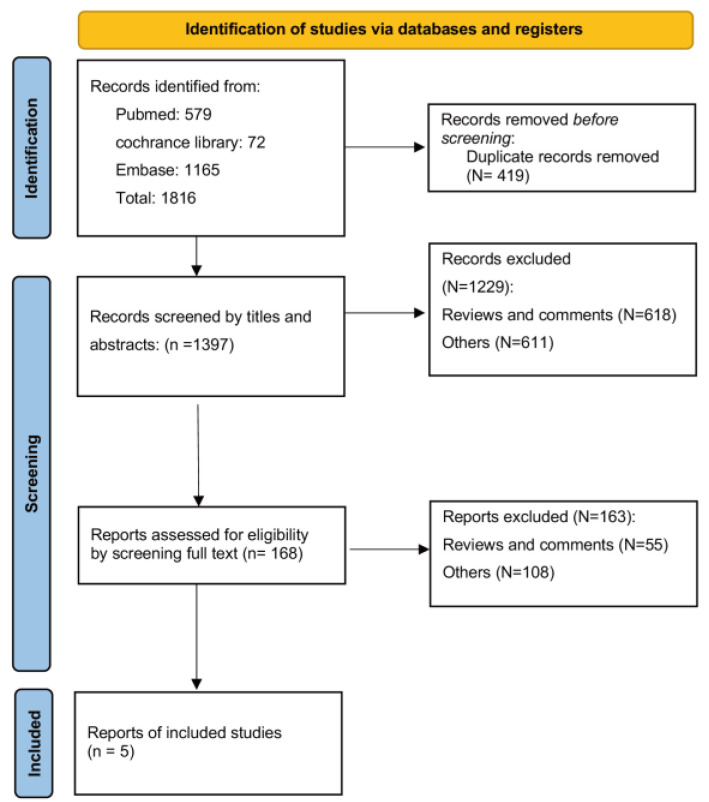
Search flow diagram.

**Table 1 nutrients-15-04462-t001:** The components of the EAT–Lancet diet.

Types of Food	Macronutrient Intake, g/Day	Caloric Intake, kcal/Day
Whole grains	232	811
Tubers or starchy vegetables	50 (0–100)	39
All vegetables	300 (200–600)	80
All fruit	200 (100–300)	126
Dairy foods		
Whole milk or derivative equivalents	250 (0–500)	153
Protein sources		
Beef, lamb		
And Pork	14 (0–28)	30
Chicken and other poultry	29 (0–58)	62
Eggs	13 (0–25)	19
Fish	28 (0–100)	40
Legumes	75 (0–100)	284
Tree nuts	50 (0–75)	291
Added fats	51.8 (20–91.8)	450
All sweeteners	31 (0–31)	120

**Table 2 nutrients-15-04462-t002:** A comprehensive summary of study characteristics.

Study, Year	Study Types	Sample, N	Mean Age	Female, %
Langmann, 2023 [27]	prospective cohort study	54,232	58.1 years	28505 (52.6%)
Zhang, 2023 [19]	prospective cohort study	24,494	Range: 50–64 year	15076 (61.5%)
Xu, 2022 [18]	prospective cohort study	59,849	55.9 years	34,512 (57.7%)
Knuppel, 2019 [31]	cross-sectional study	46 069	NA	NA
López, 2023 [17]	prospective cohort study	74,671	41.2 years	74,671 (100%)
Study, year	Database	Questionnaires	Diabetes types	Follow-up
Langmann, 2023 [27]	Danish Diet, Cancer andHealth cohort study	14-points score	Type 2 diabetes	24.3 years
Zhang, 2023 [19]	Malmö Diet and Cancer (MDC)study	42-points score	Type 2 diabetes	15.33 years
Xu, 2022 [18]	UK Biobank	14-points score	Type 2 diabetes	10 years
Knuppel, 2019 [31]	European Prospective Investigation into Cancer and Nutrition (EPIC)-Oxford study	14-points score	All types of diabetes	NA
López, 2023 [17]	The Mexican Teachers’ Cohort (MTC)	14-points score	Type 2 diabetes	2.2 years

**Table 3 nutrients-15-04462-t003:** Main findings and the quality assessments.

Study	Main Findings	The Quality Assessments
Langmann, 2023 [27]	Their results showed that greater adherence to the EAT–Lancet diet was associated with a lower risk of developing type 2 diabetes in a middle-aged Danish population.	9
Zhang, 2023 [19]	Their results demonstrated that the EAT–Lancet diet was associated with decreased risk of incident T2DM among people with different genetic risks.	9
Xu, 2022 [18]	Their results showed a higher adherence to the EAT–LDP contributes to a lower risk of T2DM	9
Knuppel, 2019 [31]	The EAT–Lancet diet showed beneficial associations for diabetes.	10
López, 2023 [17]	They found that higher adherence to the EAT–HRD score may help prevent T2D incidence among Mexican women.	9

## Data Availability

The data are available from the corresponding authors upon reasonable request.

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
