# Peer review of "The Association between the EAT–Lancet Diet and Diabetes: A Systematic Review"

_nutrients, 2023, doi:10.3390/nu15204462_

Round 1

Reviewer 1 Report

This is a systematic review article which aims to investigate the association between the EAT-Lancet diet and the risk of diabetes.

Generally, the topic is quite interesting, and the authors have in depth knowledge. They have used the appropriate methodology and the findings are sufficiently well-presented, clear, and easy to understand, so as to reach safe and solid conclusions. Overall, the manuscript is well written and structured. Thus, I think it would make a nice addition to Nutrients.

However, the following points should be generally considered, thus minor revision is demanded.

1. EAT-Lancet diet has been designed for general population, but it is also suggested for patients with diabetes. Kindly clarify.

2. Is this diet recommended by the latest guidelines? If yes, for which target population. Kindly explain.

3. Line 70-80: These findings will be further discussed in the next paragraph. Kindly add the clinical evidence on the beneficial effect of EAT-Lancet diet on other health-related parameters such as cognitive function, metabolic health (weight reduction, glucose control), cardiovascular health etc.

4. Please add numeration to the headings/subheadings.

5. Line 122: Kindly replace with the author initials.

6. Line 123-125: Patient characteristics (age, comorbidities, type and duration of diabetes, follow-up) to be added and discussed.

7. Line 142: Kindly separate Table 2 into one for main study findings and another one for quality assessment.

8. Kindly report the % of EAT diet- mediated reduction in the risk of T2DM to all the studies, if applicable.

9. Line 162: Table 2 should be placed nearby the respective paragraph.

10. Table 2 is quite confusing in the last columns presenting the results of Lopez et al. Kindly update.

11. 4 out of 5 studies indicate the beneficial effect of EAT-Lancet diet on the risk of T2D. Is there any evidence about T1D? If not, consider rewording the title in a more T2D-focused one.

12. Line 179: Figure 1 should be transitioned in another place nearby to the respective paragraph.

13. Line 183: EAT-LDI is presented for first time here. Kindly provide more information in the part of introduction or shortly explain here.

14. Information about the adherence rates, diet components, scaling system, used questionnaires would be a nice addition.

15. Line 241-242: Shortly prescribe main findings as additional health benefits.

16. Line 252-254: References to be added.

17. Line 252-254: This part needs to be further discussed to support the main findings.

Minor editing of English language is required. 

Author Response

We would like to thank your kindness for reviewing our manuscript. You have provided very detailed and highly constructive comments and thoughtful suggestions to our work, which have helped us significantly improve the manuscript. We try our best to revise our manuscript according to your constructive comments. Following are our point-by-point response. Thank you very much.

Reviewer 1:

This is a systematic review article which aims to investigate the association between the EAT-Lancet diet and the risk of diabetes.

Generally, the topic is quite interesting, and the authors have in depth knowledge. They have used the appropriate methodology and the findings are sufficiently well-presented, clear, and easy to understand, so as to reach safe and solid conclusions. Overall, the manuscript is well written and structured. Thus, I think it would make a nice addition to Nutrients.

However, the following points should be generally considered, thus minor revision is demanded.

  1. EAT-Lancet diet has been designed for general population, but it is also suggested for patients with diabetes. Kindly clarify.

Thank you very much for your suggestion, and we clarify it.

  1. Is this diet recommended by the latest guidelines? If yes, for which target population. Kindly explain.

Thank you very much for your suggestion, it seems that this diet has not been recommended by the latest guidelines since it is a new type of diet.

  1. Line 70-80: These findings will be further discussed in the next paragraph. Kindly add the clinical evidence on the beneficial effect of EAT-Lancet diet on other health-related parameters such as cognitive function, metabolic health (weight reduction, glucose control), cardiovascular health etc.

Thank you very much for your suggestion, we add a paragraph: EAT-Lancet diet has some potential beneficial effect for health-related parameters such as cognitive function, metabolic health and cardiovascular health. The nutrient-dense components of EAT-Lancet diet are essential for cognitive health, providing antioxidants and omega-3 fatty acids that combat processes impairing brain function. The diet's high fiber content stabilizes blood sugar levels, contributing to optimal metabolic health, and aids in the prevention of conditions like insulin resistance and diabetes. Moreover, it supports cardiovascular health by promoting the intake of unsaturated fats and reducing foods high in saturated fats, helping manage blood pressure, cholesterol levels, and overall heart and artery health, thereby lowering the risks of heart disease and stroke.

  1. Please add numeration to the headings/subheadings.

Thank you very much for your suggestion, and we add the numeration to the headings/subheadings.

  1. Line 122: Kindly replace with the author initials.

Thank you very much for your suggestion, and we replace them with the author initials.

  1. Line 123-125: Patient characteristics (age, comorbidities, type and duration of diabetes, follow-up) to be added and discussed.

Thank you very much for your suggestion, and we add them.

  1. Line 142: Kindly separate Table 2 into one for main study findings and another one for quality assessment.

Thank you very much for your suggestion, add we revise it.

  1. Kindly report the % of EAT diet- mediated reduction in the risk of T2DM to all the studies, if applicable.

Thank you very much for your suggestion, and we are sorry about it since it is not applicable from the included studies.

  1. Line 162: Table 2 should be placed nearby the respective paragraph.

Thank you very much for your suggestion, and we correct it.

  1. Table 2 is quite confusing in the last columns presenting the results of Lopez et al. Kindly update.

Thank you very much for your suggestion, and we correct it.

  1. 4 out of 5 studies indicate the beneficial effect of EAT-Lancet diet on the risk of T2D. Is there any evidence about T1D? If not, consider rewording the title in a more T2D-focused one.

Thank you very much for your suggestion, and we rewrite it and add it to the limitations as follows: and among these studies, 4 out of 5 studies indicate the beneficial effect of EAT-Lancet diet on the risk of types 2 diabetes, and one study demonstrated that the EAT-Lancet diet was related to the risk of diabetes. The association between EAT-Lancet diet and Type 1 diabetes should be investigated in further studies

  1. Line 179: Figure 1 should be transitioned in another place nearby to the respective paragraph.

Thank you very much for your suggestion, and we correct it.

  1. Line 183: EAT-LDI is presented for first time here. Kindly provide more information in the part of introduction or shortly explain here.

Thank you very much for your suggestion, and we add it: the EAT-LDI is ranging from 0 to 42 points, with a score of 3 to the highest adherence and a score of 0 to the lowest for 14 components in EAT-Lancet diet.

  1. Information about the adherence rates, diet components, scaling system, used questionnaires would be a nice addition.

Thank you very much for your suggestion, and we add them.

  1. Line 241-242: Shortly prescribe main findings as additional health benefits.

Thank you very much for your suggestion, and we rewrite it: for example, in a separate study by Colizzi et al., researchers devised a diet score centered on the EAT-Lancet diet and examined its connection with cardiovascular incidents and ecological footprints, analyzing data from 35,496 individuals in the EPIC-NL study. The results indicated that participants who closely followed the EAT-Lancet diet reported fewer instances of cardiovascular ailments (14% reduction), coronary artery disease (12% reduction), and strokes (11% reduction). From an environmental perspective, staunch adherence to the EAT-Lancet diet was associated with diminished greenhouse gas emissions, land utilization, freshwater and marine eutrophication, and soil acidification. A recent analysis delved into the correlation between following the EAT-Lancet diet and the occurrence and death rates of lung cancer, utilizing information from the Prostate, Lung, Colorectal, and Ovarian (PLCO) trial involving 101,755 American adults. While past studies underscored the diet's potential in diminishing chronic disease risks and overall death rates, its direct link to lung cancer remained ambiguous. The findings revealed that optimal compliance with the EAT-Lancet diet, as reflected by higher scores, correlated with a marked reduction in lung cancer cases and death rates, especially in non-small-cell lung cancer instances. This suggests that strict adherence to the diet could contribute to lowering lung cancer risks.

  1. Line 252-254: References to be added.

Thank you very much for your suggestion, and we add the references.

  1. Line 252-254: This part needs to be further discussed to support the main findings.

Thank you very much for your suggestion, and we add it: in our systematic review, all included studies demonstrated that EAT-Lancet diet was related to a lower risk of developing diabetes, especially for type 2 diabetes. The benefits may be associated with the components of EAT-Lancet diet, emphasizing fruits, vegetables, whole grains, legumes, nuts, and unsaturated oils, with minimal red meat, processed meat, and sugar, aligns with key principles for managing and preventing diabetes. Rich in high-fiber, nutrient-dense foods, it supports glycemic control, insulin sensitivity, and weight management, essential factors in diabetes care. Moreover, its low allowance for foods linked to increased diabetes risk, such as red and processed meats and refined carbohydrates, further reinforces its potential benefits.

Reviewer 2 Report

Table 1:  (a)there are some inconsistencies based to the report of the EAT-LANCET diet

https://eatforum.org/content/uploads/2019/07/EAT-Lancet_Commission_Summary_Report.pdf

(b) Under the introduction refer on the way the EAT-Lancet diet  derived to g macronutrient/d  and kcal/day

2. Maybe you need to refer to criticism of EAT-Lancet diet

3. A suggestion to re- write the Quality Appraisal (add the citation) :

Quality appraisal and data extraction were conducted using various tools depending on the study design. For randomized controlled trials (RCTs), the RoB2 tool was utilized. Non-RCTs were assessed using the ROBINS-I tool, specifically designed for such study types. Cohort and case-control studies were evaluated using the Newcastle-Ottawa Scale (NOS), which comprises eight essential questions addressing participant selection, comparability of study groups, and confirmation of exposure. Cross-sectional studies were assessed based on the criteria outlined in the AHRQ checklist, which includes 11 distinct items.

To ensure consistency in the presentation of results, scores obtained from these assessment tools were categorized into three quality levels: low, moderate, and high quality. The data extraction process was independently carried out by two authors, namely Lin and Wang. Information was systematically recorded in a predefined template, capturing study details such as the year of publication, study type, sample size (N), female percentage (%), database source, and main findings.

4. When writing a systematic review on the topic "The association between EAT–Lancet diet and diabetes," it's important to follow a structured format and provide a comprehensive overview of the research conducted in this area. Here's a suggested outline for your systematic review (make sure you followed and included all of them):

Title:

- The Association between EAT–Lancet Diet and Diabetes: A Systematic Review

Abstract:

- Provide a concise summary of the review, including the research question, methodology, key findings, and implications.

Introduction:

- Present the background and context of the study, highlighting the importance of understanding the relationship between the EAT–Lancet diet and diabetes.

- State the research question or objectives of the review.

- Define key terms and concepts.

(also what was asked for Table 1)

Methods:

- Describe the search strategy used to identify relevant studies, including databases, search terms, and inclusion/exclusion criteria.

- Detail the study selection process, including screening, eligibility criteria, and data extraction.

- Explain the quality assessment methods used for evaluating the included studies (e.g., quality assessment tools or scales).

- Clarify how data synthesis and analysis were conducted, including statistical methods if applicable.

- Address any potential bias and sources of heterogeneity.

Results:

- Present the results of the systematic review in a structured manner, organized by key themes or categories.

- Include a flowchart depicting the study selection process (PRISMA flowchart, if applicable).

- Provide a summary of the characteristics of the included studies (e.g., study design, sample size, duration).

- Describe the main findings of each study in relation to the association between the EAT–Lancet diet and diabetes.

- Consider using tables, figures, or forest plots to visually present data if appropriate.

- Discuss any patterns, trends, or inconsistencies in the results.

Discussion:

- Interpret the findings of the included studies and their implications for understanding the association between the EAT–Lancet diet and diabetes.

- Discuss the strength of the evidence, considering study quality and potential bias.

- Explore potential mechanisms or factors that may influence the observed associations.

- Compare and contrast the results with previous research in the field.

- Highlight any limitations of the review and suggest areas for future research.

- Summarize the practical implications and relevance of the findings for healthcare professionals and policymakers.

Conclusion:

- Provide a concise summary of the key findings and their significance.

- Emphasize the implications for clinical practice, public health, or policy decisions.

- Restate the research question and its relevance.

Remember to adhere to the preferred reporting guidelines for systematic reviews (e.g., PRISMA) and maintain a transparent and rigorous approach to ensure the credibility of your systematic review.

Do a minor review 

Author Response

We would like to thank your kindness for reviewing our manuscript. You have provided very detailed and highly constructive comments and thoughtful suggestions to our work, which have helped us significantly improve the manuscript. We try our best to revise our manuscript according to your constructive comments. Following are our point-by-point response. Thank you very much.

Reviewer 2:

  1. Table 1: (a)there are some inconsistencies based to the report of the EAT-LANCET diet https://eatforum.org/content/uploads/2019/07/EATLancet_Commission_Summary_Report.pdf

Thank you very much for your suggestion, we revise the Table 1 according to this report.

(b) Under the introduction refer on the way the EAT-Lancet diet derived to g macronutrient/d and kcal/day.

Thank you very much for your suggestion, and we add: the specific macronutrient distribution is roughly around 50-55% carbohydrates, 29-35% fats, and 15-25% proteins of daily total caloric intake of around 2,500 kcal/day for men and 2,000 kcal/day for women.

  1. Maybe you need to refer to criticism of EAT-Lancet diet.

Thank you very much for your suggestion, and we add the criticism: while the EAT-Lancet diet has been praised for its potential environmental sustainability and health benefits, it has also faced criticism. Critics argue that its one-size-fits-all approach lacks adaptation to individual nutritional needs, cultural and regional food preferences, and agricultural systems, potentially making it impractical or unsustainable for certain populations. Some experts worry that the recommended reduction in meat and dairy intake might lead to deficiencies in essential nutrients like vitamin B12, iron, and calcium, particularly where alternative sources are not readily available or affordable. Additionally, the diet's emphasis on plant-based foods raises concerns about the bioavailability of certain nutrients, meaning they might not be as readily absorbed from plant sources as they are from animal products.

  1. A suggestion to re- write the Quality Appraisal (add the citation): Quality appraisal and data extraction were conducted using various tools depending on the study design. For randomized controlled trials (RCTs), the RoB2 tool was utilized. Non-RCTs were assessed using the ROBINS-I tool, specifically designed for such study types. Cohort and case-control studies were evaluated using the Newcastle-Ottawa Scale (NOS), which comprises eight essential questions addressing participant selection, comparability of study groups, and confirmation of exposure. Cross-sectional studies were assessed based on the criteria outlined in the AHRQ checklist, which includes 11 distinct items. To ensure consistency in the presentation of results, scores obtained from these assessment tools were categorized into three quality levels: low, moderate, and high quality. The data extraction process was independently carried out by two authors, namely Lin and Wang. Information was systematically recorded in a predefined template, capturing study details such as the year of publication, study type, sample size (N), female percentage (%), database source, and main findings.

Thank you very much for your suggestion, and we rewrite it:

  1. When writing a systematic review on the topic "The association between EAT– Lancet diet and diabetes," it's important to follow a structured format and provide a comprehensive overview of the research conducted in this area. Here's a suggested outline for your systematic review (make sure you followed and included all of them):

Title: - The Association between EAT–Lancet Diet and Diabetes: A Systematic Review Abstract: - Provide a concise summary of the review, including the research question, methodology, key findings, and implications.

Introduction: - Present the background and context of the study, highlighting the importance of understanding the relationship between the EAT–Lancet diet and diabetes. - State the research question or objectives of the review. - Define key terms and concepts. (also what was asked for Table 1) Methods: - Describe the search strategy used to identify relevant studies, including databases, search terms, and inclusion/exclusion criteria. - Detail the study selection process, including screening, eligibility criteria, and data extraction. - Explain the quality assessment methods used for evaluating the included studies (e.g., quality assessment tools or scales). - Clarify how data synthesis and analysis were conducted, including statistical methods if applicable. - Address any potential bias and sources of heterogeneity. Results: - Present the results of the systematic review in a structured manner, organized by key themes or categories. - Include a flowchart depicting the study selection process (PRISMA flowchart, if applicable).

- Provide a summary of the characteristics of the included studies (e.g., study design, sample size, duration). - Describe the main findings of each study in relation to the association between the EAT–Lancet diet and diabetes. - Consider using tables, figures, or forest plots to visually present data if appropriate. - Discuss any patterns, trends, or inconsistencies in the results. Discussion: - Interpret the findings of the included studies and their implications for understanding the association between the EAT–Lancet diet and diabetes. - Discuss the strength of the evidence, considering study quality and potential bias. - Explore potential mechanisms or factors that may influence the observed associations. - Compare and contrast the results with previous research in the field. - Highlight any limitations of the review and suggest areas for future research. - Summarize the practical implications and relevance of the findings for healthcare professionals and policymakers. Conclusion: - Provide a concise summary of the key findings and their significance. - Emphasize the implications for clinical practice, public health, or policy decisions. - Restate the research question and its relevance. Remember to adhere to the preferred reporting guidelines for systematic reviews (e.g., PRISMA) and maintain a transparent and rigorous approach to ensure the credibility of your systematic review.

Thank you very much for your advice, and we rewrite our systematic review according to your constructive comments.

Reviewer 3 Report

This paper was addressed to review the existing clinical studies concerning the relationship between diabetes and EAT-Lancet diet. In this regard, 5 studies of four prospective studies and one cross-sectional study and 259315 subjects were considered. On the whole, all these studies showed that adhesion to the EAT-Lancet diet implies a low risk of diabetes.

Introduction deals with the management strategies for diabetes including dietary approaches among which the EAT-Lancet diet has gained, starting from 2019, increasing consideration. Methods explain how the present study was carried out in the databases of Embase, PubMed, and Cochrane up to August 15, 2023. Moreover, inclusion criteria, quality appraisal and data extraction were defined. Results consider patient characteristics, the assessment of quality, and the relationship between EAT-Lancet diet and diabetes by analysing, in particular, the studies of Langmann et al., Zhang et al. and López et al. Three tables and one figure are enclosed resulting to have been prepared with accuracy. Discussion reports how the EAT-Lancet diet appears to represent an emergent universal health reference diet in a series of pathologies including, apart from diabetes, cardiovascular diseases, stroke, and cancer. Authors conclude that, in spite of some systematic limitations of their review, the EAT-Lancet diet may be thought to constitute a significant dietary intervention for diabetes although more high-quality studies with strict design, long-term intervention and follow-up are needed in order to further support this conclusion.

Overall, the present review is somewhat interesting as well as the dietary approach to diabetes mellitus is concerned. Lexicon, sentence fluency, “English style”, manuscript structuring, tables/figure and their legends, and references are acceptable. Some minor editorial refinements, here and there, would be desirable.

Lexicon, sentence fluency, “English style”, manuscript structuring, tables/figure and their legends, and references are acceptable.

Author Response

This paper was addressed to review the existing clinical studies concerning the relationship between diabetes and EAT-Lancet diet. In this regard, 5 studies of four prospective studies and one cross-sectional study and 259315 subjects were considered. On the whole, all these studies showed that adhesion to the EAT-Lancet diet implies a low risk of diabetes.

Introduction deals with the management strategies for diabetes including dietary approaches among which the EAT-Lancet diet has gained, starting from 2019, increasing consideration. Methods explain how the present study was carried out in the databases of Embase, PubMed, and Cochrane up to August 15, 2023. Moreover, inclusion criteria, quality appraisal and data extraction were defined. Results consider patient characteristics, the assessment of quality, and the relationship between EAT-Lancet diet and diabetes by analysing, in particular, the studies of Langmann et al., Zhang et al. and López et al. Three tables and one figure are enclosed resulting to have been prepared with accuracy. Discussion reports how the EAT-Lancet diet appears to represent an emergent universal health reference diet in a series of pathologies including, apart from diabetes, cardiovascular diseases, stroke, and cancer. Authors conclude that, in spite of some systematic limitations of their review, the EAT-Lancet diet may be thought to constitute a significant dietary intervention for diabetes although more high-quality studies with strict design, long-term intervention and follow-up are needed in order to further support this conclusion.

Overall, the present review is somewhat interesting as well as the dietary approach to diabetes mellitus is concerned. Lexicon, sentence fluency, “English style”, manuscript structuring, tables/figure and their legends, and references are acceptable. Some minor editorial refinements, here and there, would be desirable.

We would like to thank your kindness for reviewing our manuscript. You have provided highly constructive comments and thoughtful suggestions to our work, which have helped us significantly improve the manuscript, and we try our best to make editorial refinements and revise the manuscript.